# Prevalence of Antimicrobial Prescribing in Long-Term Care Facilities in a Local Health Authority of Northern Italy

**DOI:** 10.3390/ijerph192013412

**Published:** 2022-10-17

**Authors:** Andrea Sarro, Francesco Di Nardo, Michela Andreoletti, Chiara Airoldi, Lorenza Scotti, Massimiliano Panella

**Affiliations:** 1Department of Translational Medicine, Università Degli Studi del Piemonte Orientale, 28100 Novara, Italy; 2Presidio Ospedaliero Ss. Trinità, ASL NO, 28021 Borgomanero, Italy

**Keywords:** long-term facilities, antibiotic consumption, antibiotics, antimicrobial stewardship

## Abstract

Background: Almost half of antimicrobial prescriptions in long-term care facilities (LTCFs) is inappropriate. This broad use might represent a strong contributor to antimicrobial resistance in these facilities. This study aimed to assess antibiotic use patterns and potential associated factors with a survey of LTCFs in the local health authority (LHA) of Novara. Methods: A cross-sectional study was conducted in 25 LTCFs in the LHA of Novara following the healthcare-associated infection in LCTFs (HALT) protocol. Information on residents and facilities was assessed. Antibiotic usage and potential determinants were also estimated. Results: In total, 1137 patients were screened for antibiotic usage. Mean age was 84.58 years (SD 9.77), and the majority were female (76.52%). Twenty-six were antibiotic users (prevalence rate 2.29%, 95%CI 1.50–3.33). Antimicrobials were mainly prescribed orally (84.62%). Potential risk factors for antibiotic prescription were catheter use (central and peripheral venous, *p*-values 0.0475 and 0.0034, respectively, and urinary, *p*-value 0.0008), immobilization (*p*-value < 0.0001), and sex (*p*-value 0.0486). Conclusions: This study identified a low prevalence of antimicrobic consumption in LTCFs. Further surveillance studies are warranted to identify trends and changes in pathogen incidence and antimicrobial resistance and to inform public health authorities on the necessity of prudent use of antimicrobials in LCTFs.

## 1. Introduction

In recent years, in most European countries, a significant increase in life expectancy in the population aged 65 years has been registered, leading to an ever-growing and fast rise in the request for elderly healthcare services, such as home care, nursing homes, and healthcare facilities [1]. These facilities provide and deliver a blend of residential care to people unable to live independently, requiring assistance in daily living activities, and needing less intensive medical care than that provided in hospitals [2].

According to a recent survey performed by the Italian National Statistical Institute (ISTAT), in Italy, the elderly represents 23.5% of the entire population [3]. Regarding care facilities, 12,828 social welfare and social health residential structures with 390,689 beds were available and distributed throughout the territories of the Italian health sanitary districts (HSDs) on 31 December 2015. In those facilities, sanitary assistance was provided to 382,634 people, of whom almost 288,000 (75.2%) were aged over 65 years [4].

The population projections performed by the EU Centre for Disease Prevention and Control (ECDC) estimate a certain increase in the number of residents in long-term care in almost all other European countries in future decades, in line with the previously described situation in Italy. Specifically, by 2050, the number of people aged over 65 per 100 people of working age will reach 50%, and the population aged 80 years and over is projected to increase from 16.8 million (4.1%) in 2010 to 43.3 million (11.5%) in 2060 [5].

Patients living in healthcare facilities are usually frail, vulnerable, and at high risk of infection due to their age, age-related health problems, and also because of living in closed facilities in strict contact with other residents [6].

Immunosenescence, comorbidities, and malnutrition—pathological conditions very common among elderly patients—contribute largely to the burden of healthcare-related infections (HAIs), which represent an important issue and concern in long-term care facilities (LTCFs) due to their high prevalence and incidence [7].

The most commonly reported infections are urinary tract infections (UTIs), lower respiratory tract infections, skin infections, and gastroenteritis, among others [8].

HAIs are listed among the main causes of antibiotic prescribing. A recent systematic literature review showed that antibiotic usage in LCTFs is common, and its annual prevalence rates range from 47% to 79% each year [9]. Almost half of the antibiotics prescribed in LTCFs are not appropriate [10]; this recurrent use of antibiotics represents an important driver and one of the strong contributors to antimicrobial resistance. Consequently, LTCFs, similar to hospitals or ambulatory care, may represent the focus for multidrug-resistant bacteria, leading to frequent hospitalizations and rehospitalizations [11,12].

Antimicrobial resistance represents a significant public health concern in LCTFs due to its high mortality and impact on the health system, causing several patient safety issues [13]. Recent projections on the incidence and prevalence of multidrug-resistant microorganisms showed a steadily increasing trend in the European area. In support of this, it has been estimated that every year, antibiotic-resistant bacteria cause about 20% of the diagnosed infectious diseases globally [14].

Antimicrobial stewardship in nursing homes is a recent but essential concept created to limit the further development of resistance; strategies settled in acute care settings may be adapted to the specific LTCF context [15].

The first step toward improved antimicrobial prescribing is to analyze the current patterns of antimicrobial use.

To quantify them, the ECDC has started surveillance, ushering in an effective project known as healthcare-associated infection in LCTFs (HALT) in 2010, a point prevalence survey aimed to provide a deeper insight into antibiotic prescribing patterns in European LTCFs [16].

Italy was included in the 2013 HALT-2 and in the 2017 HALT-3 studies [17,18].

Quantifying the prevalence of antibiotic prescribing and consumption in LCTFs is warranted to better understand the determinants and the magnitude of this phenomenon, informing local health authorities and administrators on their prudent use.

Surveys might represent a helpful tool in the fields of epidemiology and public health to give an overall picture of antimicrobial use in LCTFs, due to minimal resources within those facilities that aim to conduct constant surveillance of antibiotic prescribing [19].

Different from previous similar investigations mostly conducted in entire nations and in facilities with mixed populations, the current study aimed to bring particularly focused evidence on this issue.

Our aim was to assess the prevalence and the potential factors associated with antibiotic usage through a point prevalence survey on residents in LTCFs in the territories of a small administrative entity (LHA of Novara) in order to guide local stewardship programs on the appropriate use of antibiotics and on the need for guidelines available on this theme in LCTF protocols.

## 2. Materials and Methods

### 2.1. Study Design and Source Population

A cross-sectional study was conducted on LTCFs in the LHA of the Province of Novara. This survey was conducted and initiated under the supervision and control of the healthcare-associated infection control team of the LHA of Novara (Piedmont Region, Novara, Italy), including towns and municipalities of different sizes with a population of 428,535 as of 31 December 2019.

An LHA is an administrative territorial entity of the National Health Service in Italy introduced by the National Health Service Reorganization Act of 1992 [20].

The LHA of Novara is divided into three health sanitary districts: (i) Novara city, (ii) south area, and (iii) north area. In the territory of the LHA of Novara, there are 42 LTCFs.

A map of Novara province is shown in Figure 1, reporting boundaries and the LCTFs distributed across the territories of health sanitary districts.

An LTCF could be defined as an institution in which medical doctors’ and nurses’ care is guaranteed to patients 24 h a day.

All LTCFs belonging to the Novara LHA were invited to join the study, and all residents who were present on the day of the survey were eligible for inclusion.

Residents were defined as people who were living in an LTCF for at least 48 h before the day of the survey. Residents who were not living full-time in the LTCF (e.g., residents from daycare centers) or living full-time in the LTCF but who were not present on the day of the survey (e.g., absent for hospitalization) were excluded from the study.

Data were collected through a questionnaire administered to the LHCs’ personnel between the 20 and 27 February 2022; no direct involvement of patients was required.

The study was revised by the ethical committee of the LHA of Novara. All procedures conducted in this study followed the ethical standards of the institutional and/or national research committee and with the 1964 Helsinki Declaration and its later amendments or comparable ethical standards.

### 2.2. Questionnaire and Data Collection

The questionnaire used for data collection was an adapted version of the HALT protocol of the ECDC [21] LTCFs personnel (directors, chief nurses, and medical doctors previously informed about the aims and the methods of the survey) were mandated to complete it, supported by local researchers and survey coordinators who ensured and guaranteed its correct compilation. First of all, information regarding the LTCFs was collected, such as the number of available and occupied beds on the days of data collection and the number of nurses, physicians, and medical assistants working in the structure. Then, for each resident, the following characteristics were collected: (i) demographic characteristics (age, sex), (ii) care load indicators (fecal and/or urinary incontinence, dementia, impaired mobility, bedsores, and other wounds), (iii) clinical characteristics (use of wheelchair, use of nasogastric probe, use of urinary catheter, bedridden patient, use of central venous catheter, use of peripheral venous catheter, dialysis), (iv) hospitalization within 48 h before the interview, (v) previous surgery in the 30–90 days before the interview, and (vi) data on antimicrobial use. All residents were screened for HAIs and antibiotic usage. Further information was collected on subjects undergoing antimicrobial treatment. These included the diagnostic tests performed to identify and determine the infection, infection site, agent, type of therapy (empirical, targeted, preventive), active ingredient, compound name, indication for therapy, prescribed doses, route of administration, and treatment duration. The indication for antibiotic use, infectious diseases, and causative agents/antibiotics were defined in accordance with ECDC protocol [21]; no information was collected on antiviral and antifungal use.

### 2.3. Statistical Analysis

Descriptive statistics were used to summarize LTCF and resident characteristics. Continuous variables were reported as mean and standard deviation (SD) or median and interquartile range (IQR) if not normally distributed, and categorical variables as absolute frequencies and percentages. The prevalence of antibiotic use was estimated as the ratio between the number of antibiotic users and the number of subjects living in the LTCF who were present on the day of the survey. Exact 95% confidence intervals (95%CIs) were also calculated. Prevalence, overall and stratified by district, was estimated. Chi-squared test, Fisher exact test, or *t*-test (continuous variables) were also used in order to evaluate the potential association between LTCF residents’ demographic and clinical characteristics and the probability of antibiotic usage. The prevalence of antibiotic use and corresponding 95%CI was also calculated, stratified by the variables associated with antibiotic use, to better appreciate the difference between patients’ characteristics.

The individual address records and the position of LCTFs in Figure 1 were geocoded as Universal Transverse Mercator (UTM) geographic coordinates, and latitude and longitude coordinates were recorded. Moreover, the geographical coordinates of Novara and its LHA boundaries were obtained and derived from the ISTAT records, considering the district as the unit of interest.

Tests were two-tailed, and the type I error was fixed at 0.05. All statistical analyses herein reported were performed using SAS v 9.4 (SAS Institute, Cary, CA, USA) and R v 3.4.1 (R Core Team, Vienna, Austria).

## 3. Results

Overall, 25 out of the 42 LCTFs invited to take part in the study actually provided the requested information, corresponding to 1137 residents. LCTF characteristics are reported in Table 1.

Most LCTFs included in our study were located in the north area district and were almost of similar size, with a median number of available and occupied beds of 58 (IQR 35–65) and 41 (IQR 33–57), respectively. We only report on two LCTFs with fewer than 10 patients. Residents in these facilities were followed by a median number of 16 medical assistants and a median number of 4 nurses; 1 physician was available for all LCTFs. The median age of the residents included in our analysis was 84.85, and the majority were female.

The prevalence of antibiotic use and the corresponding exact 95%CI, overall and stratified by district, is reported in Figure 2.

Of the 1137 residents, 26 were antibiotic users and received at least one antimicrobial agent on the day of the survey; therefore, we revealed a corresponding prevalence rate of 2.29% (95%CI 1.50–3.33). No significant differences were observed between the three sanitary districts of the Novara LHA; specifically, the prevalence-specific estimates varied from 2.14% for the Novara city district to 2.60% for the southern area.

Table 2 reports the HAI and antibiotic use characteristics for the 26 antibiotic users included in our analysis.

The most common HAI was respiratory tract infection, with a relative frequency of 59.09% (n = 13), followed by urinary tract infection (n = 4, 18.18%). Skin/wound infections, surgical site infections, gastrointestinal tract infections, and genital tract infections were observed equally in 4.55% of residents on antimicrobial treatments on the day of the survey. The lack of antimicrobial results for HAI testing concerned the majority of HAIs (62.5, n = 5), as a diagnostic test was performed for only 37.5% of HAIs.

The sputum was the commonest sample and diagnostic material used to diagnose the infection. All the residents on antimicrobial treatment were dispensed one antibiotic on the day of the survey. None of them received more than one antimicrobial agent. Antimicrobials were mainly prescribed orally. Indeed, 84.62% of residents on antimicrobial treatment received antimicrobial treatment per oral administration, while the parental route was used for only four patients (intravenously and intra-muscularly). None of the LCTF residents received antibiotics per nasogastric tube or percutaneous gastric tube; in addition, a nasal or rectal administration route was not reported. In addition, antimicrobial agents were mainly prescribed within the same LCTFs.

The most reported indication for the prescribed antimicrobials was for treatment (84%); only 12% of residents received treatment with antibiotics for prophylaxis. The indication was missing and not reported for just one subject. According to the patient charts, antibiotics were mainly prescribed empirically (64%, n = 16), and only five residents (20%) received targeted therapeutic antibiotics. The median treatment duration was 6 days. Penicillins with beta-lactamase inhibitors and macrolides represented the most prescribed group of antibacterial (63.56% of all prescriptions). Within those groups, specifically, amoxicillin/clavulanate (n = 5) was the most prescribed antibiotic, followed by azithromycin. Fluoroquinolones (levofloxacin and ciprofloxacin) were the third most prescribed antibiotics (three subjects). Antibacterials for systemic use accounted for 15% of all antimicrobial prescriptions. Six out of eight residents who received antibiotics, with available information on the agent, were colonized with SARS-CoV-2, and only one with Klebsiella pneumoniae.

The distribution of LCTF residents’ clinical and demographic characteristics is reported in Table 3.

A total of 97 subjects (1 antibiotic user and 96 nonusers) had missing values for all patient demographic and clinical characteristics. Among the remaining 1040 subjects (25 antibiotic users and 1015 nonusers), the mean age of included subjects was 84.58 years (SD 9.77), and the majority were female (76.52%). Disorientation and dementia were present in 502 residents (50.91%). Pressure sores affected 5.97% of residents; moreover, 630 patients (60.58%) were in a wheelchair, and 2.60% were immobilized and bedridden. Few residents had a nasogastric probe, and four were on dialysis treatment; 6.63% of residents needed a urinary catheter, while 2.89% needed a peripheral intravenous one. Finally, nine residents had previous surgery (0.87a), and only 0.87% were hospitalized within the 48 h prior to the days of the survey.

Only a few factors were associated with antibiotic usage: catheter use (central and peripheral venous, *p*-values 0.0475 and 0.0034, respectively, and urinary, *p*-value 0.0008), immobilization (*p*-value < 0.0001), and sex (*p*-value 0.0486). Specifically, subjects using catheters had a higher probability of being antibiotic users and immobilized patients, while females were less frequent antibiotic users. For variables associated with antibiotic use, the prevalence of antibiotic users and the corresponding 95%CI for each stratum are reported in Table 4.

## 4. Discussion

This multicenter cross-sectional study evaluated antimicrobial consumption and prescribing among the 1137 residents of 25 LTCFs distributed across a small province sited in Northern Italy, providing a comprehensive view of antibiotic consumption in LCTFs distributed across the territories of the sanitary districts of Novara LHA.

Our analysis highlighted a prevalence rate of antibiotic prescribing of 2.2% on the day of the survey, which turned out to be an unexpected finding. Indeed, a previous internal survey conducted 3 years ago with a similar methodology unveiled a higher consumption of antibiotics in LCTFs in Novara LHA.

Quantification of the usage of antibiotics in LCTFs, necessary to inform public health policies on prudent and appropriate antimicrobial use, was analyzed in some observational cross-sectional studies, such as the previously cited HALT studies [16,17,18] and Finnish studies [22].

Different from our investigation, they were mostly conducted in vast geographic areas (e.g., entire nations) [23,24].

To our knowledge, a focus on small administrative entities, such as a single LHA, was less frequent.

The current survey revealed antibiotic consumption in LCTFs in the territories of Novara LHA below the prevalence estimated by a narrative review published recently (3–11%) and 2.6% lower than the prevalence rate that emerged from the most recent survey performed in 24 European countries and 2 European candidate countries [25,26].

The first study on antibiotics usage in LCTFs was performed by the European Surveillance of Antimicrobial Consumption (ESAC), and this included 323 LCTFs distributed across the territories of all EU countries. The main prevalence rate of antibiotic prescribing calculated was 6.3% (range 1–17.3%) [27]. Then, in the wake of this point prevalence analysis, ECDC conducted three studies in 2010, 2013, and 2017 with similar designs and methods. The prevalence rates of antimicrobial usage that emerged were respectively 4.3% (HALT-1) (range 0.0–13.3%), 4.4% (HALT-2) (range 1–12.1%), and (HALT-3) [14,17].

Italy also participated in the aforementioned HALT-2 and HALT3; the antibiotic usage found was 4% and 3.9%, respectively, higher than the prevalence rate registered in our study.

Nevertheless, the findings that emerged here are not entirely comparable due to the several typologies of LCTFs included in the other studies. To analyze the phenomenon in a homogenous sample and reduce the risk of selection bias, we decided to exclude LCTFs with mixed populations. Indeed, we included only LCTFs that deliver residential care to elderly people (>65 years) unable to live independently and requiring assistance in daily living activities. In addition, discrepancies in comparing all those surveys could be partially justified by the different data collection times. We decided to perform our cross-sectional survey at the end of February, when the peak of flu was overcome, in order to reduce the risk of a potential bias.

Residents included in our study do not differ significantly compared with previous European surveys [14,17].

We found a median age of 84.36 among patients in treatment with antibiotics, in line with other similar studies conducted in Slovenia (median age 83.5) and the rest of Europe (median age 85) [23]. In support of those findings, a study conducted with the same methodology in Finland on the consumption of antibiotics in LCTFs during a study period of 1 year showed that patients with age ≥80 present a higher risk of assuming antibiotics [22].

Most of the antibiotic users included in our analysis were male, in line with a previous cross-sectional Canadian survey [28]. Indeed, a higher prevalence of antibiotic consumption was found in male patients (p = 4.10, 95%CI 4.07–4.13).

By contrast, the Finnish and Swiss studies previously mentioned showed a specific influence of the female sex on the prevalence of the consumption of antibiotics [22,29]. The HALT studies did not identify any positive correlation between sex and the prevalence of antibiotics usage.

Forty-eight percent of our patients on antibiotic treatment were diagnosed with dementia. Nevertheless, unlike the Slovenian study, we did not find a higher prevalence of dementia diagnosis among patients on antimicrobial treatment [23]. Conversely, in the Finnish study, antibiotic therapy was more frequent in patients with cognitive disorders [22].

In accordance with the Finnish study, we also found a positive association between being immobilized and higher usage of antibiotics [22].

The most prescribed antibiotics were amoxicillin plus clavulanate and macrolides, partially in line with European data.

Amoxicillin and clavulanate were prescribed more frequently than the other penicillins.

In addition, a worrying concern is the high usage of fluoroquinolones, which, similar to amoxicillin, is listed among the broad-spectrum antibiotics responsible for antibiotic resistance [30].

Among patients on treatment with antibiotics, only 12% received them for prophylaxis, which is lower than in other European studies. HALT2 and HALT3 registered remarkable prophylactic usage of antimicrobial drugs (respectively 27.7 and 27.2). The prevalence of residents on treatment with antimicrobial agents for prophylaxis varies across countries. The aforementioned Slovenian point prevalence survey demonstrated recourse to prophylaxis for only 1.2% of enrolled residents [23]. By contrast, in the Swiss study, 23.5% of the antibiotics were prescribed for prophylaxis [29].

The aim of this investigation was not to assess the appropriateness of antibiotic prescribing and usage in LCTFs in Novara LHA. A recently published survey conducted in the Netherlands demonstrated that improvements in appropriateness might result in a drastic and significant reduction in antibiotic prescription prevalence [31]. However, we registered antimicrobial agent usage that was lower than the HAIs diagnosed. This could be interpreted as a proxy for the appropriateness of antibiotic prescribing and usage in terms of guideline indications. Adherence to guidelines might contribute in a significant way to reducing inappropriate prescriptions of antibiotics [32]. Therefore, the development and implementation of more specific guidelines for antibiotic usage in LTCFs, which are currently scarce and vague, should be encouraged and appear necessary.

Lastly, we found frequent usage of azithromycin in patients with COVID-19. Azithromycin is listed among the broad-spectrum antimicrobial agents with good absorption after oral intake and powered by a long half-life [33]. General practitioners widely use this antibiotic in Italy for the therapeutic management of COVID-19 due to its well-known antiviral activity in bronchial epithelial cells and its anti-inflammatory and immunoregulatory effects. The results that emerged from a recently published systematic review and meta-analysis did not support the usage of azithromycin, which we registered in patients with SARS-CoV-2 infections included in our sample [34].

### Strengths and Limitations

This study presents some limitations that must be taken into account. Firstly, this study was carried out only on the population of the Novara LHA, which represents a small part of the Italian population. Indeed, our results may not be entirely generalizable to the entire nation. Secondly, data collected are part of a point prevalence study, and, due to this, they offered only a photograph at that point in time.

Another limitation of this survey was its cross-sectional design; a survey, especially if administered on only 1 day, could be potentially subjected to several variations. Nevertheless, we decided to adopt this methodological approach because of its feasibility when applied to settings characterized by limited human resources in terms of the workforce, such as LCTFs, especially during the COVID-19 pandemic.

On the other hand, among the strengths of our study, it is possible to cite the use of a standardized survey across all participating LCTFs, the collection of detailed data on LCTF characteristics, and the use, when necessary, of administrative health databases.

A broad number of LCTFs responded to our invitation, so the large sample size of the population included provides a detailed picture of antimicrobial consumption in LCTFs. In addition, the baseline demographic and clinical characteristics of the included population, similar to several studies from European countries, suggest that our data are not influenced by potential selection bias and offer a realistic photograph.

## 5. Conclusions

Our results might represent a useful tool to better understand the potential determinants and the magnitude of this issue and to inform local and national public health policymakers on the importance of the prudent prescription of antimicrobial agents.

Similar routine surveillance studies are warranted to identify trends and changes in pathogen incidence and antimicrobial resistance, including new pathogens at local, national, and global levels.

Periodic monitoring of antibiotic consumption through standardized systems in line with national and European legislation is also needed to build and clarify approaches aimed at controlling antimicrobial resistance, guiding clinicians’ and policymakers’ decisions on the appropriate use of antibiotics and on the need for adopting and incorporating guidelines available on this theme in LCTF protocols.

## Figures and Tables

**Figure 1 ijerph-19-13412-f001:**
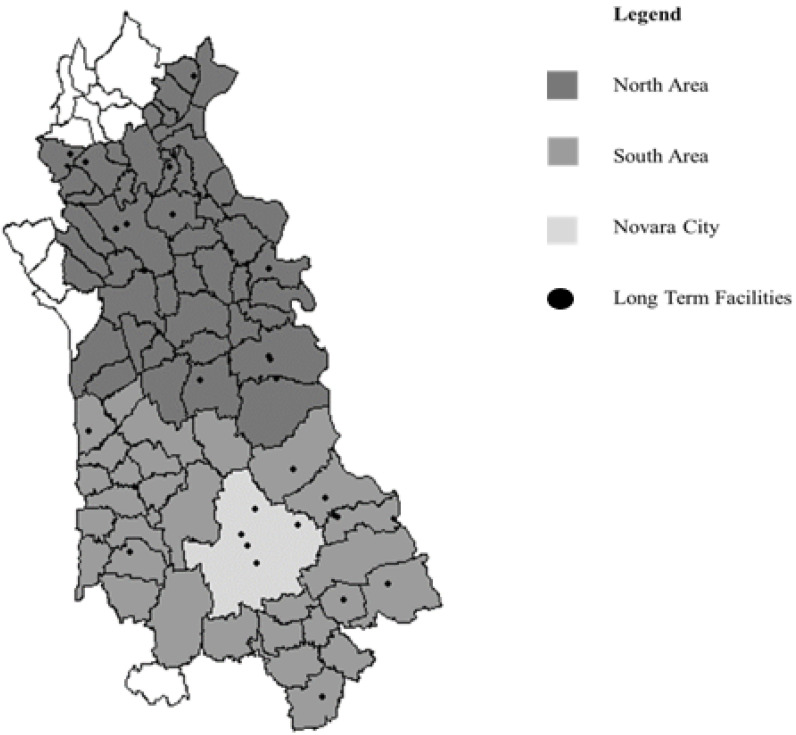
Map of Novara LHA. Colors represent the health sanitary districts, while long-term facilities are reported as black dots.

**Figure 2 ijerph-19-13412-f002:**
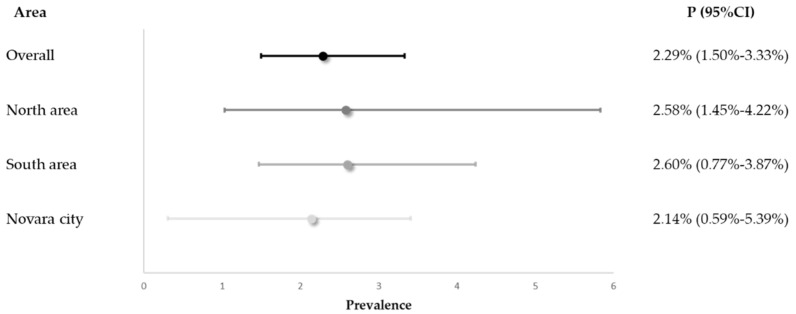
Prevalence of antibiotic use (P) and exact 95% confidence intervals (95%CI), overall and by district.

**Table 1 ijerph-19-13412-t001:** Descriptive statistics of LCTF characteristics.

	N = 25
**District, N (%)**	
*North area*	14 (56%)
*South area*	8 (32%)
*Novara city*	3 (12%)
**N° of available beds, median (IQR)**	58 (35–65)
**N° of occupied beds, median (IQR)**	41 (33–57)
**N° medical assistants, median (IQR)**	16 (13–20)
**N° of nurses, median (IQR)**	4 (3–5)
**N° of physicians, median (IQR)**	1 (1–1)

**Table 2 ijerph-19-13412-t002:** Characteristics of antibiotic use and the test used to identify infections.

	Antibiotic Users N = 26
	**N (%)**
**Infection site**	
*Others*	1 (4.55)
*Skin/wound*	1 (4.55)
*Surgical site*	1 (4.55)
*Gastrointestinal tract*	1 (4.55)
*Genital tract*	1 (4.55)
*Respiratory tract*	13 (59.09)
*Urinary tract*	4 (18.18)
*Missing*	4
**Test to evaluate infection**	
*No*	15 (62.50)
*Yes*	9 (37.50)
*Missing*	2
**Test material**	
*Expectoration*	6 (66.67)
*Feces/recap swab*	1 (11.11)
*Urine*	2 (22.22)
*Missing*	17
**Agent**	
*Klebsiella pneumoniae*	1 (12.50)
*SARS-CoV-2*	6 (75.00)
*Others*	1 (12.50)
*Missing*	18
**Compound**	
*Amoxicillin and beta-lactamase inhibitor*	5 (22.73)
*Azithromycin*	4 (18.18)
*Cefixime*	1 (4.55)
*Ceftriaxone*	1 (4.55)
*Ciprofloxacin*	1 (4.55)
*Clarithromycin*	1 (4.55)
*Fosfomycin*	1 (4.55)
*Gentamicin*	1 (4.55)
*Levofloxacin*	2 (9.09)
*Teicoplanin*	1 (4.55)
*Sulfamethoxazole and trimethoprim*	1 (4.55)
*Vancomycin*	1 (4.55)
*Missing*	6
**Route of administration**	
*Intravenous*	2 (7.69)
*Intramuscular*	2 (7.69)
*Oral*	22 (84.62)
**Therapy type**	
*Other*	1 (4.00)
*Empirical*	16 (64.00)
*Targeted*	5 (20.00)
*Preventive*	3 (12.00)
*Missing*	1
**Treatment duration (days)**	6 (5–8)

**Table 3 ijerph-19-13412-t003:** Distribution of LCTF residents’ clinical and demographic characteristics and the *p*-value of the test used to evaluate the relationship between patients’ characteristics and antibiotic use.

Variable	Antibiotic Use			
	No N = 1015	Yes N = 25	Total N = 1040	*Chi-Squared Test* *p*-Value
	**N (%)**	**N (%)**	**N (%)**	
**Sex**				
*Males*	234 (23.08)	10 (40.00)	244 (23.48)	0.0486
*Females*	780 (76.92)	15 (60.00)	795 (76.52)	
*Missing*	1	0	1	
**Age, mean (SD)**	84.59 (9.82)	84.36 (7.46)	84.58 (9.77)	0.9159 ^
**Urinary incontinence**				
*No*	252 (23.84)	4 (16.00)	256 (24.62)	0.3114
*Yes*	763 (75.17)	21 (84.00)	784 (75.38)	
**Bowel incontinence**				
*No*	419 (41.28)	9 (36.00)	428 (41.15)	0.5861
*Yes*	596 (58.72)	16 (64.00)	612 (58.85)	
**Dementia**				
*No*	471 (49.01)	13 (52.00)	484 (49.09)	0.7679
*Yes*	490 (50.99)	12 (48.00)	502 (50.91)	
*Missing*	54	0	54	
**Pressure sores**				
*No*	955 (94.18)	22 (88.00)	977 (94.03)	0.1974
*Yes*	59 (5.82)	3 (12.00)	62 (5.97)	
*Missing*	1	0	1	
**Other wounds**				
*No*	976 (96.16)	22 (88.00)	998 (95.96)	0.0759 *
*Yes*	39 (3.84)	3 (12.00)	42 (4.04)	
**Wheelchair**				
*No*	396 (39.01)	14 (56.00)	410 (39.42)	0.0860
*Yes*	619 (60.99)	11 (44.00)	630 (60.58)	
**Immobilized patient**				
*No*	993 (97.93)	17 (68.00)	1010 (97.40)	<0.0001 *
*Yes*	21 (2.07)	6 (24.00)	27 (2.60)	
*Missing*	1	2	3	
**Nasogastric probe**				
*No*	1009 (99.41)	25 (10.00)	1034 (99.42)	1.0000*
*Yes*	6 (0.59)	0 (0.00)	6 (0.58)	
**Central venous catheter**				
*No*	1014 (99.90)	24 (96.00)	1038 (99.81)	0.0475 *
*Yes*	1 (0.10)	1 (4.00)	2 (0.19)	
**Peripheral venous catheter**				
*No*	988 (97.44)	19 (82.61)	1007 (97.11)	0.0034 *
*Yes*	26 (2.56)	4 (17.39)	30 (2.89)	
*Missing*	1	2	3	
**Urinary catheter**				
*No*	953 (93.89)	18 (72.00)	971 (93.37)	0.0008 *
*Yes*	62 (6.11)	7 (28.00)	69 (6.63)	
**Dialysis**				
*No*	1011 (99.61)	25 (10.00)	1036 (99.62)	1.0000 *
*Yes*	4 (0.39)	0 (0.00)	4 (0.38)	
**Hospitalization (48 h before the interview)**				
*No*	1009 (99.41)	24 (96.00)	1033 (99.33)	0.1570 *
*Yes*	6 (0.59)	1 (4.00)	7 (0.67)	
**Previous surgery (30–90 days before the interview)**				
*No*	1005 (99.21)	24 (96.00)	1029 (99.13)	0.1977 *
*Yes*	8 (0.79)	1 (4.00)	9 (0.87)	
*Missing*	2	0	2	
**Infection or colonization**				
*No infection/colonization*	953 (95.78)	5 (20.00)	958 (93.92)	<0.0001 *
*Colonization*	2 (0.20)	0 (0.00)	2 (0.20)	
*Probable infection/colonization*	32 (3.22)	11 (44.00)	43 (4.22)	
*Confirmed infection/colonization*	8 (0.80)	9 (36.00)	17 (1.67)	
*Missing*	20	0	20	

^ *t*-test, * Fisher exact test.

**Table 4 ijerph-19-13412-t004:** Prevalence and corresponding confidence intervals for the association between the study covariables and antibiotic consumption in the study population.

	Antibiotic Users	Total		
	**N = 26**	**N = 1040**		
	**N**	**N**	**Prevalence (p)**	**95%CI**
**Sex**				
*Males*	10	244	4.10%	4.07–4.13
*Females*	15	795	1.89%	1.88–1.89
**Immobilized patient**				
*No*	17	1010	1.68%	1.68–1.69
*Yes*	2	27	7.41%	6.91–7.91
**Central venous catheter**				
*No*	24	1038	2.31%	2.31–2.32
*Yes*	1	2	50.00%	25.5–74.50
**Peripheral venous catheter**				
*No*	19	1007	1.89%	1.88–1.89
*Yes*	4	30	13.33%	12.58–14.09
**Urinary catheter**				
*No*	18	971	1.85%	1.85–1.86
*Yes*	7	69	10.14%	9.89–10.40

A higher prevalence of antibiotic use was observed in males (p = 4.10%, 95%CI 4.07–4.13), immobilized residents (p = 7.41%, 95%CI 6.91–7.91), and in patients in treatment with central and peripheral venous (p = 50.00%, 95%CI 25.5–74.50 and p = 13.33%, 95%CI 12.58–14.09, respectively) and urinary catheter (p = 10.14%, 95%CI 9.89–10.40).

## Data Availability

The datasets generated or analyzed during this study are available and can be obtained, at request, from Andrea Sarro (e-mail: andrea.sarro@uniupo.it).

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
