# Peer review of "Prevalence of Antimicrobial Prescribing in Long-Term Care Facilities in a Local Health Authority of Northern Italy"

_ijerph, 2022, doi:10.3390/ijerph192013412_

Round 1

Reviewer 1 Report

The manuscript is an interesting, althouhg limited, picture (in time and space) of the prevalence of antimicrobial prescribing in long-term care facilities in Northern Italy. However, the limitations are soundly adressed and the manuscript provides some intresting results and conclusion on this matter in a small context in Northern Italy, providing information that might be used to confront similar case studies.

Reviewer 2 Report

The authors conducted a survey for antimicrobial usage in Long Term-Care Facilities of the Local Health Authority of Novara, Italy, however, a lot of similar surveys were conducted in large populations as the authors also cited in this study. I suggest the authors point out the significance of this study compared with other works, and explain how this work is unique and how this work can help other researchers with their studies.

“The current study aimed to assess the prevalence and the potential factors associated with antibiotic usage through a point-prevalence survey on the residents in LTCFs in the territories of the LHA of Novara, providing a deeper insight into the need for implementation of preventive programs of antimicrobial stewardship, also identifying possible gaps in current knowledge.” (Row 85-89, Page 2). The statistical population in this study is too small to support this statement. I suggest that the authors make a clearer explanation and connection between the prevalence and the potential factors associated with antibiotic usage.

Author Response

Please the attachment

Reviewer 3 Report

First of all I want to congratulate the authors for conducting this study and accepting its limitations.

My concern is related to results section. In text (Line183,188), you mention 26 residents that were antibiotic users, but in Table 3, although N=26 for antibiotic users, in subsections it looks like some of the patients are missing information (Males 10, Females 15). I recommend you revise and correct the whole table regarding Total number of patients (N=1040, but sums in subsections vary between 986 - 1041), number of patients using antibiotics (N=26, but sums in subsections vary between 19 to 25) , number of patients not using antibiotics (N=1014 and sums vary between 961 to1015). Also, please verify the statistical analysis with new data. If the data in the table is correct and explained by line 224 ("105  subjects were excluded from the analysis due to missing values in all patients"), then I would recommend more explanations on this topic and add for each subsection "Missing info" - number of patients line. 

In line 247 you mention you analyze 1141 residents, while the whole study refers to 1040 residents (Table 3) and in Methods section you mention 1137 residents (Line 168). Also 105 subjects were excluded from the study (Line 224). Please correct and explain the design of the study. Maybe a chart is useful.

Conclusion section is a bit long and I personally see some discussions there (Line362-363, 372-373). 

Author Response

Please the attachment.

Reviewer 4 Report

It is a very well written work, with an appropriate methodology,
well supported from a bibliographic point of view.
Only 2 points need to be clarified: 1) In Table 2, Prednisone, which does not belong to the pharmacological
class of antibiotics, appears among the antibiotics used by patients:
please explain better why it was included; 2) In lines 324-326, a very interesting point is mentioned:
the data you found, lower than the European average and the rest of Italy,
can be given by the awareness on the issue of the correct use of
probiotics. However, there is no bibliographic reference on previous
guidelines, please expand this part.

In general Compliments

Round 2

Reviewer 3 Report

The accuracy of data improved significantly. Great job! Congratulations